# Treatment Approaches and Outcome of Patients with Neuroendocrine Neoplasia Grade 3 in German Real-World Clinical Practice

**DOI:** 10.3390/cancers14112718

**Published:** 2022-05-31

**Authors:** Simone Luecke, Christian Fottner, Harald Lahner, Henning Jann, Dominik Zolnowski, Detlef Quietzsch, Patricia Grabowski, Birgit Cremer, Sebastian Maasberg, Ulrich-Frank Pape, Hans-Helge Mueller, Thomas Matthias Gress, Anja Rinke

**Affiliations:** 1UKGM Marburg, Department of Gastroenterology, Philipps University Marburg, 35037 Marburg, Germany; simone.luecke@gmx.de (S.L.); gress@med.uni-marburg.de (T.M.G.); 2Department of Internal Medicine I, Endocrinology, University Hospital Mainz, 55131 Mainz, Germany; christian.fottner@unimedizin-mainz.de; 3Department of Endocrinology and Metabolism, University Hospital of Essen, 45147 Essen, Germany; harald.lahner@uk-essen.de; 4Department of Gastroenterology and Hepatology, Campus Virchow Klinikum, University Medicine Charité, 10117 Berlin, Germany; henning.jann@charite.de; 5Department of Oncology, Klinikum Chemnitz, 09116 Chemnitz, Germany; d.zolnowski@skc.de; 6Praxis Dr. med. habil. Diener, 09376 Oelsnitz/Erzgebirge, Germany; bd.quietzsch@t-online.de; 7Klinikum Havelhöhe, Campus Virchow Klinikum, Institute of Medical Immunology, MVZ Oncology, University Medicine Charité, 10117 Berlin, Germany; patricia.grabowski@charite.de; 8Department of Oncology, University Hospital of Cologne, 50923 Cologne, Germany; birgit.cremer@uk-koeln.de; 9Department of Internal Medicine and Gastroenterology, Asklepios Klinik St. Georg, 20099 Hamburg, Germany; s.maasberg@asklepios.com (S.M.); ul.pape@asklepios.com (U.-F.P.); 10Institute of Medical Biometry and Epidemiology, Philipps University Marburg, 35037 Marburg, Germany; hans-helge.mueller@staff.uni-marburg.de; 11German NET Registry, German Society of Endocrinology, 90518 Altorf, Germany

**Keywords:** neuroendocrine neoplasia, neuroendocrine tumor G3, neuroendocrine carcinoma, Ki67, chemotherapy, prognosis

## Abstract

**Simple Summary:**

Grade 3 neuroendocrine neoplasms (NEN G3) are a rare and heterogeneous subtype of NEN and include poorly differentiated neuroendocrine carcinomas and well-differentiated neuroendocrine tumors G3 (NET G3). Standard chemotherapy with platinum plus etoposide may not be appropriate for all subgroups, but more tailored approaches suffer from the lack of data. In our study, we provide real-world data from a large center-based cohort of the German NET Registry and hope to stimulate efforts to conduct clinical trials for well-defined entities.

**Abstract:**

Background: Neuroendocrine neoplasia grade 3 (NEN G3) represents a rare and heterogeneous cancer type with a poor prognosis. The aim of our study was to analyze real-world data from the German NET Registry with a focus on therapeutic and prognostic aspects. Methods: NEN G3 patients were identified within the German NET Registry. Demographic data and data on treatments and outcomes were retrieved. Univariate analyses were performed using the Kaplan–Meier-method. Multivariate analysis was performed using a Cox proportional hazard model. Results: Of 445 included patients, 318 (71.5%) were diagnosed at stage IV. Well-differentiated morphology (NET G3) was described in 31.7%, 60% of cases were classified as neuroendocrine carcinoma (NEC), and the median Ki67 value was 50%. First-line treatment comprised chemotherapy in 43.8%, with differences in the choice of regimen with regard to NET or NEC, and surgery in 41.6% of patients. Median overall survival for the entire cohort was 31 months. Stage, performance status and Ki67 were significant prognostic factors in multivariate analysis. Conclusions: The survival data of our national registry compare favorably to population-based data, probably mainly because of a relatively low median Ki67 of 50%. Nevertheless, the best first- and second-line approaches for specific subgroups remain unclear, and an international effort to fill these gaps is needed.

## 1. Introduction

Grade 3 neuroendocrine neoplasms of gastroenteropancreatic origin (GEP NEN G3) are rare cancers accounting for less than 1% of all gastrointestinal malignancies [1] and 5–15% of gastroenteropancreatic neuroendocrine neoplasms [2,3,4,5]. Due to the lack of specific clinical trials in patients with GEP NEN G3, treatment recommendations are mostly derived in analogy to small cell lung cancer (SCLC). Since the WHO classification of 2010, GEP NEN G3 has been defined by the proliferation marker Ki67 with a value above 20% [6]. Increasing evidence of the clinical, molecular and prognostic heterogeneity of the G3 cohort led to the introduction of a new NET G3 category in the 2017 WHO classification for pancreatic NEN [7] and later also for gastrointestinal (GI) NEN in the 2019 WHO classification [8] to distinguish morphologically well-differentiated neuroendocrine tumors G3 (NET G3) from poorly differentiated neuroendocrine carcinoma (NEC G3). In lung NEN, NET G3 has not been defined yet, but it is included within atypical carcinoids that can have a mitotic count exceeding 10/mm^2^ and/or a proliferation rate above 20% [9].

The prognosis of NEC patients is poor, with a median overall survival (OS) of 10 months, as has been shown in an analysis of the Surveillance, Epidemiology, and End Results (SEER) Program for the whole cohort of poorly and undifferentiated NEN [10], and 6.4 months for patients with distant disease [11]. Most data on NET G3 patients derived from specialized NEN centers demonstrated a longer survival of 40–99 months [12,13,14,15].

The Nordic NEC trial reported different response rates and outcomes by dividing the cohort at the cut-off level of a Ki67 rate of 55%: whereas the subgroup of highly proliferative NEN G3 (Ki67 ≥ 55%) showed a response rate (RR) of 42% to platinum-based chemotherapy but a poor prognosis with a median OS of 10 months, the subgroup with Ki67 < 55% demonstrated limited response to this chemotherapy with an RR of 15% but a better OS of 14 months [16].

Cis- or carboplatin in combination with etoposide has been used for decades in NEC patients [17,18] and is still the recommended first-line treatment in international guidelines [1,19,20]. Second-line options mainly encompass other chemotherapeutic regimens that have limited efficacy [21,22,23,24,25,26].

The best therapeutic strategy for NET G3 patients and NEN G3 with Ki67 < 55% is largely unknown. Often, platinum-based chemotherapy has been used as first-line treatment, but considering the limited response rates, other options such as temozolomide-based chemotherapy [27,28,29], FOLFOX chemotherapy [29,30] and other strategies belonging to the treatment repertoire for NET G2 patients, such as everolimus [31], sunitinib [32] and peptide receptor radionuclide therapy (PRRT) [33], have been suggested.

In NEN G3 patients, surgery is usually not recommended in advanced disease [1,20].

The aim of our study was to analyze real-world data from a large nationwide cohort of NEN G3 patients from the German NET Registry. The main focus was to evaluate prognoses with regard to histological (morphology and proliferation rate), clinical (gender, age, performance status, stage and primary tumor localization) and biochemical (tumor markers and lactate dehydrogenase levels) parameters. We also wanted to characterize therapeutic strategies used for patients with neuroendocrine tumors and neuroendocrine carcinomas.

## 2. Materials and Methods

The German NET Registry is a multicentric and multidisciplinary project of currently 57 actively participating institutions in Germany caring for patients with NEN; it is organized by the German Society of Endocrinology (Deutsche Gesellschaft für Endokrinologie, DGE, current President: Prof. Dr. G. Stalla, Munich, Germany). The German Registry was founded in 2004; the structure of the initial version has previously been described [34]. In brief, at the participating centers, prior to the documentation of patient data, signed informed consent was obtained from NEN patients considered for recruitment. Further inclusion criteria comprise: histologically confirmed NEN of the gastroenteropancreatic (GEP) system, of the lung or of unknown primary tumor localization and an age at the time of recruitment to the NET Registry of at least 18 years.

For the period between 1999 and 2004, data were documented retrospectively and thereafter prospectively. Documented data were transferred to an MS ACCESS database (Lohmann & Birkner Health Care Consulting, GmbH, Berlin, Germany).

Specific inclusion criteria for the current study were histologically proven NEN G3, defined as NEN with a Ki67 value above 20%, or documented neuroendocrine carcinoma according to the 2010, 2017 or 2019 WHO classifications. Specific exclusion criteria were: small cell lung cancer, known primary outside the gastroenteropancreatic system and the lung (e.g., gynecological or urological neuroendocrine carcinoma) or mixed histologies (mixed adeno-neuroendocrine carcinoma/MANEC; mixed neuroendocrine/non-neuroendocrine neoplasms/MiNEN).

The collected data included personal data such as gender, age, date of initial diagnosis, date of diagnosis of NEN G3, body mass index (BMI) and performance status at the time of diagnosis, second malignancy, last visit or date and—if available—cause of death. Disease-specific information, such as primary tumor localization, presence or absence of metastasis, date of detection of metastasis, localization of metastasis, presence or absence of functionality, available histopathological classification criteria (neuroendocrine tumor or neuroendocrine carcinoma and Ki67) and staging information, was also obtained. If more than one histological report was available, the highest documented Ki67 value was used for further analyses. Finally, treatment-specific information on treatment modalities and results with regard to overall outcomes were also recorded. S.L. visited the centers to update information on treatment and survival until 8/20, when personal visits had to be stopped due to the COVID pandemic.

The leading ethics committee is the Ethic Committee at Charité Mitte (Berlin, Germany), and the updated web-based version was approved on 24 October 2013 (EA1/279/13). In addition, approval by the local ethics committee is mandatory for every participating center.

### Statistical Analysis

Median overall survival (OS) was calculated from the time point of the first diagnosis of NEN G3 (date of histology) to the date of death or the date of the last presentation at the center. Time to treatment failure was defined as the interval between the start of a treatment and the termination of the treatment due to progression, side effects, patient wishes or death. Univariate analysis of overall survival was performed using the Kaplan–Meier method and tested for the significance of differences using log-rank testing. Multivariate analysis of potentially independent prognostic factors was performed using a Cox proportional hazard model. A *p*-value of < 0.05 was considered to be statistically significant. Descriptive data are reported either as numbers, percentages or medians with their ranges.

Statistical evaluation was performed using R Studio 1.3.1093 (R Studio PBC, 2020, Boston, MA, USA), R 4.0.3 (2020, The R Foundation for Statistical Computing) and SAS (SAS Institute, Cary, NC, USA). 

## 3. Results

### 3.1. Description of the Cohort

A total of 445 patients with histologically confirmed NEN G3 with an initial diagnosis between 2000 and 6/2020 were included in the analysis.

Query of the German NET Registry database for “NEN G3”, Ki67 > 20% and “neuroendocrine carcinoma” resulted in 637 cases. We excluded 59 patients with primaries outside the GEP system and the lung and 43 patients with mixed histologies (MINEN (mixed neuroendocrine/non-neuroendocrine neoplasms)/MANEC (mixed adeno-neuroendocrine carcinoma)). In 58 patients, NEN G3 could not be confirmed, as only Ki67 values up to 20% were documented, Ki67 was missing or stainings for both neuroendocrine markers chromogranin A and synaptophysin were negative. A review of the data for double inclusion identified 8 patients, and in 24 patients, no follow-up information was available. Therefore, finally, 445 patients could be included for further analyses. Figure 1 summarizes the patient selection process.

The median age at diagnosis of the 246 male (55.3%) and 199 female (44.7%) patients was 63 years. More than 60% of patients had a good performance status (ECOG 0 and 1) at diagnosis, and 26.6% initially presented without distant metastases. Only a minority of 26 patients (5.8%) had functioning tumors (including carcinoid syndrome, insulinoma, gastrinoma, glucagonoma and VIPoma). The most common primary sites were the pancreas (29.7%), unknown (CUP 21.8%), and the colon and rectum (19.1%). The most common metastatic sites were the liver (67.4%), lymph nodes (55.3%) and bone (21.3%). At first diagnosis, histology was NET G1/2 in 11.9% of cases, progressing to NEN G3 during the course of the disease. Overall, tumors were classified as NET in 31.7% and NEC in 60%, and morphological classification was missing in 8.3%. Median Ki67 was 50%. In 49 patients (11.1%), second malignancies were documented; the most common were prostate cancer (*n* = 7), skin tumors (*n* = 6) and breast cancer (*n* = 5). Patient characteristics are summarized in Table 1.

### 3.2. Treatment

The median number of treatment lines was two (average 2.9). A total of 10.3% of patients received more than 5 lines of treatment, and a maximum of 15 subsequent treatment lines were documented in one patient. Six patients did not receive tumor treatments.

#### 3.2.1. First-Line Treatment

Surgery was the first treatment modality in 185 patients (41.6%), with similar rates in patients diagnosed with NET G3 (42.3%) and NEC (41.6%). The majority of patients with locoregional disease at diagnosis (85/118; 72%) underwent resection as the first treatment approach, but in patients with stage IV disease, surgery was also the second most common first-line treatment (95/318; 29.9%). The primary tumor in stage IV patients was more often resected in patients with colorectal and intestinal NEN (25/58; 43.1% and 9/24; 37.5%, respectively) compared to patients with pancreatic (27/103; 26.2%) or pulmonary primaries (5/18; 27.8%).

Systemic chemotherapy was the most frequently applied first-line treatment (195 patients; 43.8%). Cis- or carboplatin plus etoposide was the most frequently used first-line protocol, given in 102 NEC patients (74.5%) and 11 NET G3 patients (26.2%). Fourteen NET G3 patients (33.3%) received a combination of temozolomide and capecitabine, six received FOLFOX (14.2%), and five NET G3 patients (11.9%) received streptozocin and 5FU as first-line chemotherapy, respectively. Other first line treatments comprised somatostatin analogs (*n* = 33, 7.4%), local radiotherapy (*n* = 8, 1.8%), everolimus (*n* = 7; 1.6%) and PRRT (*n* = 6; 1.4%).

With respect to first-line chemotherapy (also including patients who were first surgically treated), cis- or carboplatin plus etoposide was the most often chosen regimen, given in 165 NEC patients (72%) and 30 NET G3 patients (33%). The combination of temozolomide and capecitabine was similarly often used in NET G3 patients (*n* = 29; 31.9%), followed by streptozocin and 5FU (*n* = 12; 13.2%). These protocols were less often used in NEC patients (*n* = 15, 6.6% and *n* = 3, 1.3%, respectively), in whom other platinum-containing schedules such as FOLFOX or carboplatin plus paclitaxel were the second most frequently used (*n* = 31; 13.5%). 

Figure 2 demonstrates the most frequently selected first-line therapies in patients with NEC and NET G3.

#### 3.2.2. Second-Line Treatment

Chemotherapy was the most frequently selected treatment modality in 203 of 332 patients (61.1%) who received second-line treatment (74.6% of the total cohort), and 37% and 49% of the NET G3 subgroup and NEC subgroup, respectively, were treated with a second-line chemotherapy. More than 40 different protocols were given as second-line chemotherapy, most often cis- or carboplatin + etoposide (*n* = 101), temozolomide + capecitabine (*n* = 22), FOLFOX (*n* = 22), FOLFIRI (*n* = 10) and streptozocin and 5FU (*n* = 8). Other therapies comprised surgery (*n* = 46; 13.9%), radiotherapy (*n* = 28; 8.4%), PRRT (*n* = 19; 5.7%), SSA (*n* = 17; 5.1%), everolimus (*n* = 9; 2.7%), liver embolization (*n* = 7; 2.1%), ablative treatments (*n* = 2; 0.6%) and sunitinib (*n* = 1; 0.3%). 

#### 3.2.3. Higher Lines of Treatment

A total of 213 patients (47.9%) received at least three treatment lines, including different chemotherapy protocols in 110 patients. Other treatment modalities comprised: surgery in 25 patients (11.7%), radiotherapy in 21 patients (9.9%), targeted treatments in 19 patients (8.9%), PRRT in 18 patients (8.5%), locoregional treatments of liver metastases in 11 patients (5.2%) and SSA in 9 patients (4.2%). 

#### 3.2.4. Treatment and Outcome

The median survival in patients without treatment was 4 months (range 0–10).

In metastatic disease, there was a non-significant trend toward longer mOS in NEC patients who received cisplatin + etoposide (*n* = 39; mOS 18 months) as first-line treatment compared to those treated with carboplatin + etoposide (*n* = 44; mOS 14 months; *p* = 0.2).

With respect to first-line systemic chemotherapy in patients classified as NET G3, outcomes seemed to be slightly better with first-line STZ/5FU (*n* = 12; only cases of pancreatic NET G3, mOS not reached) and temozolomide/capecitabine (*n* = 29; mOS 50 months) compared to cisplatin + etoposide (*n* = 17; mOS 44 months), carboplatin + etoposide (*n* = 13; mOS 46 months) and FOLFOX (*n* = 9; mOS 21 months), without reaching statistical significance (*p* = 0.06).

In patients with NEN with a proliferation rate (Ki67) below 55%, the median time to treatment failure (mTTF) was shorter when patients received first-line cis- or carboplatin plus etoposide (5 months) compared to temozolomide plus capecitabine (9 months) and other platin-containing chemotherapies such as FOLFOX (9 months) and streptozocin plus 5FU (12 months) (*p* = 0.005).

The majority of patients with cancers with Ki67 ≥ 55% received cis- or carboplatin plus etoposide as first-line treatment, with a mTTF of 5 months. There were no statistically significant differences in mTTF when other options were used (*p* = 0.1). However, this may be due to the low numbers of other first-line treatments. 

With a duration of 5 months, the mTTF of second-line treatments was generally short. 

For NEC patients, second-line treatment with temozolomide + capecitabine and carboplatin + etoposide resulted in the longest mTTFs of 8 and 6 months, respectively, without statistically significant differences (*p* = 0.2). In patients with NET G3, a favorable mTTF in second-line treatments was documented for streptozocin + 5FU (16 months), temozolomide + capecitabine (12 months) and PRRT (12 months), whereas FOLFIRI and platinum-based chemotherapy were less effective, with mTTFs of 2 to 4 months (*p* = 0.001). 

### 3.3. Survival: Putative Prognostic Factors

#### 3.3.1. Overall Survival in the Entire Cohort

During a median observation period of 15 months (0–298), 210 patients died. The median overall survival (OS) for all NEN G3 in the registry was 31 months. The overall cohort demonstrated 1-, 2- and 5- year survival rates (YSRs) of 76.1%, 58.1% and 32.8%, respectively.

#### 3.3.2. Univariate Analyses

The possible prognostic roles of the baseline characteristics age at diagnosis, gender, performance status, body mass index (BMI), stage, primary tumor localization, differentiation, proliferation rate (Ki67) and levels of CgA, LDH and NSE were analyzed.

Although women had a slightly longer overall survival of 34 months compared to men, with a median OS of 28 months, this was not statistically different (*p* = 0.2).

In contrast, increasing age at diagnosis was associated with poorer outcomes (HR 1.013 per year, *p* = 0.0182).

Performance status was clearly related to survival. Whereas patients with an ECOG PS of 0 at diagnosis survived a median of 38 months, the median OS in patients with ECOG PS 1 and ECOG PS 2–4 was 24 months and 15 months, respectively (Figure 3a).

There was a trend toward longer mOS with increasing BMI at diagnosis (available in 289 cases) without statistically significant differences.

Patients with locoregional disease displayed a longer overall survival compared to patients with distant metastases at the time of diagnosis (mOS 112 months, 88 months and 24 months in patients with localized disease, stage III and stage IV, respectively, *p* = 0.00182) (Figure 3b).

We could not demonstrate a prognostic influence for the primary tumor localization in the entire cohort. In the subgroup of patients with distant metastases, patients with a pancreatic primary had a better prognosis, with a mOS of 30 months compared to only 10 months in patients with colorectal primaries and 20 months in CUP (*p* = 0.01) (Figure 3c).

The median OS in patients with cancer classified as NEC was 26 months and shorter compared to NET G3 (mOS 44 months, *p* = 0.01) (Figure 3d). This was also true for patients initially diagnosed with metastatic disease: mOS in stage IV NET G3 and NEC was 34 months and 20 months, respectively (*p* = 0.01). 

The 53 patients initially diagnosed as NET G1/G2 and evolved to NET G3 during the course of the disease survived a median of 203 months (0.95 CI: 73-na) from the first diagnosis, whereas mOS from the time of diagnosis of G3 was only 34 months and not statistically longer compared to patients initially diagnosed with NEN G3 (mOS 31 months; *p* = 0.6). 

The proliferation rate (Ki67) was prognostically relevant. The Kaplan–Meier plot with the most frequently used cut-off value of 55% is shown in Figure 3e. In our cohort, the analysis using a median Ki67 cut-off value of 50% improved the statistical significance between these two proliferation groups: mOS in patients with Ki67 < 50% was 46 months compared to 20 months in patients with Ki67 ≥ 50%, *p* = 0.000335 (Figure 3f). The 2 and 5Y survival rates in these subgroups were 75.2% and 44.2% for patients with Ki67 < 50% and 42.2% and 22.3% for patients with Ki67 ≥50%, respectively. In patients with distant metastases at diagnosis and a proliferation rate of at least 50%, only 26.2% and 12% survived more than 2 and 5 years, respectively.

As demonstrated in Figure 3g,h, elevated baseline values for LDH and CgA were prognostically unfavorable. A normal level of LDH was associated with longer OS (41 months), whereas mOS was 18, 16 and 2 months in patients with LDH levels elevated < twice the upper limit of normal, 2-< 5 times the upper limit of normal and >5 times the upper limit of normal, respectively (*p* = 0.015). NSE was available in 138 cases at baseline, and only marked elevations >10-fold the upper limit of normal were associated with poor outcomes (mOS 9 months). 

#### 3.3.3. Multivariate Analysis

Table 2 summarizes the results of a multivariate analysis, which demonstrated an independent prognostic role for stage (distant metastases at diagnosis versus locoregional disease), the proliferation rate (Ki67) (continuous covariate) and the ECOG performance status (0 versus 1–4).

Due to a correlation of the ECOG PS with age, age as a continuous covariate became significant after the exclusion of ECOG PS. The same was true for Ki67 and morphological classification.

## 4. Discussion

In this study, we analyzed real-world data from a large multicenter cohort of patients suffering from high-grade neuroendocrine tumors mainly of gastrointestinal origin. Due to the rarity and heterogeneity of these prognostically unfavorable cancers, the best therapeutic strategy is unclear. In addition, the subgroup of NET G3 was introduced rather recently in the 2017 and 2019 WHO classifications; therefore, prospective data for patients with morphologically well-differentiated but highly proliferative neuroendocrine tumors are lacking. Publications on this topic have mostly been limited to small retrospective, often monocentric series on treatment outcomes [21,22,24,31,35,36,37] or larger population-based registry data lacking relevant information such as the proliferation marker Ki67 and information about systemic treatments [38]. 

The comparison of results in publications on high-grade NEN is hampered by the use of different inclusion criteria. Whereas the recent, population-based registry publications [38,39] define high-grade as poorly differentiated without considering the proliferation rate (Ki67), the NORDIC NEC trial [16] and several other retrospective treatment trials [21,31,32,35] used the 2010 WHO classification, defining NEN G3 as Ki67 above 20% but not differentiating between NET G3 and NEC. Some reports include all primaries [38], and others concentrate on gastrointestinal origin [16,39]. Our cohort includes patients with all stages at diagnosis in the same way as population-based publications, whereas series on treatment outcomes usually focus on advanced disease. The majority of patients are diagnosed with metastatic disease, 71.5% in our study, which is comparable to 69.3% of patients in the SEER publication with all primaries included [38] and even slightly higher than in the GI NEC cancer registry publication (64.6%) [39]. The liver is the most frequent metastatic site, whereas brain metastases are rare in patients with extrapulmonary NEN G3. Of our entire cohort of patients, 8.5% were diagnosed with cerebral metastases, but only 4.6% of the patients were diagnosed with GI primary. This is in accordance with the results from the Nordic NEC publication (4% brain metastases) [16] and confirms that prophylactic brain irradiation is not indicated [1]. A median age of 63 years and a slight male preponderance is in line with other publications [16,39,40]. The vast majority of NEN G3 cases were non-functioning, and only 5.8% of our patients suffered from a hormonal syndrome. Functionality is more often present in NET G3 compared to NEC patients [12,15,41], which is supported by our data and a very recent publication on NET G3 with a higher rate of functionality (11.3%) [40]. Performance status (PS) at the time of diagnosis was generally good in our multicenter cohort, with 48% PS 0 and 41.1% PS 1 whenever documented (available in 69.4% of cases). These figures are more favorable compared to the cohort of the Nordic NEC trial, with 28% PS 0 and 46% PS 1 cases [16]; in most other publications, information on PS is lacking. NET G3 cases accounted for 31.7% of our entire cohort and may be overrepresented compared to other multicenter publications, in which the rate of NET G3 is between 12.3% [42] and 18.1% [12]. This is also reflected by a relatively low median Ki67 of 50% (median Ki67 30% in NET G3 and 70% in NEC), whereas in the NORDIC NEC cohort, 53% of patients had Ki 67 values ≥ 55%. 

Several publications have demonstrated that patients with morphologically well-differentiated NEN G3 have a better prognosis compared to NEC [12,13,14,15,41,43], which also led to the introduction of the NET G3 category in the latest WHO classifications [7,8]. In univariate analysis, mOS was significantly longer in NET G3 than in NEC patients (44 months versus 26 months, *p* = 0.01); however, with the inclusion of the proliferation rate (Ki67) in the multivariate analysis, the difference was no longer statistically significant. In contrast, the Ki67 value was also confirmed as a prognostic parameter at the multivariate level. In the initial NORDIC NEC trial, patients with Ki67 ≥ 55% had a poorer mOS of 11 months compared to 14 months in patients with Ki67 < 55%, but the proliferation rate was not a statistically independent prognostic parameter in multivariate analysis [16]. Interestingly, and in contrast to our data, the recent re-evaluation of 196 patients of the Nordic NEC study demonstrated that the Ki67 values were not prognostic in either NET G3 or NEC patients [42]. Patients classified as NEC with Ki67 < 55% had the same poor prognosis of a mOS of 11 months as NEC with Ki67 ≥ 55%, whereas the outcome in NET G3 was more favorable, with a mOS of 33 months. Several other publications also demonstrated the prognostic relevance of Ki67 in high-grade NEN [33,44,45], and it was suggested as one of five parameters for a prognostic score of GI NEC [46].

Not surprisingly, stage was an important prognostic factor in our study, which is in line with other publications [12,38,42,46]. PS turned out to be of independent prognostic relevance, which was also shown in the NORDIC NEC trial [16,42] and was proposed to be included in a prognostic score for GI NEC [46]. In population-based data, the information on performance status is often missing, but the confirmed prognostic relevance emphasizes that this simple parameter should always be included in oncological trials. 

An elevation of lactate dehydrogenase (LDH) levels was associated with poor survival in patients with NEN G3 [16,46,47,48]. This is supported by our data demonstrating a longer survival in patients with normal LDH levels at diagnosis in univariate analysis (excluded from multivariate analysis due to a high number of missing values). LDH levels act as an indicator of tumor burden and aggressiveness in many types of cancer and have been demonstrated to be a poor outcome parameter in patients with colorectal cancer [49], pancreatic cancer [50], ovarian cancer [51], non-small cell lung cancer (NSCLC) [52] and small cell lung cancer [53], a common neuroendocrine carcinoma. 

There are conflicting results regarding the role of the primary tumor location in outcome. Pancreatic primary was associated with poor survival of only 5.7 and 6 months in American cancer registry analyses [38,39]. In a large Japanese retrospective trial, the outcome of pancreatic patients (mOS 8.5 months) was similar to that of patients with colorectal primaries (mOS 7.6 months) [48] and primary tumor localization without prognostic relevance in a French retrospective multicenter NEC trial [47], whereas in the NORDIC NEC trial [16] and also some smaller retrospective treatment trials [35], patients with pancreatic primaries survived longer compared to patients with colorectal primary. We could not demonstrate the prognostic relevance of primary tumor localization for the entire cohort, but in line with the NORDIC NEC trial, in patients with metastatic disease, pancreatic primary was associated with a favorable outcome (mOS 30 months) compared to colorectal primary (mOS 10 months) (Figure 3c). This can mainly be explained by a high proportion of NET G3 in the pancreatic subgroup, in contrast to the high proportion of NEC in the colorectal subgroup. The different inclusion of NET and NEC patients is probably one reason for the conflicting results, although in the update of the Nordic NEC trial in the subgroup of NEC patients (without NET G3), pancreatic primary was associated with better outcomes in a multivariate analysis [42]. 

Overall, the outcome in our multicenter cohort, with a mOS of 31 months, compares favorably with data from population-based [38,39] or other multicenter-based registries [16,47]. This is partly because of the large proportion of the prognostically more favorable NET G3 subgroup with a correlating low median Ki67. As mentioned above, the median OS of the (mostly metastatic) NET G3 cohort in the update of the NORDIC NEC trial was 33 months [42] and thus very similar to the mOS of 34 months in our metastatic NET G3 subgroup. The Spanish center-based registry included 364 NEN G3 cases and reported 5-year survival rates of 35.3% and 21.9% in the subgroups of Ki67 ≤ 50% and > 50%, respectively [43]. The corresponding figures of our cohort were slightly better for the subgroup with Ki67 below 50% (44.2%) and nearly identical for the subgroup with proliferation rates above 50% (22.3%). 

The subgroup classified as NEC in our study clearly demonstrated better outcomes than expected from population-based registries. Whether this is due to a more tailored and interdisciplinary approach used in specialized NEN centers or whether it reflects the bias that fitter patients are more likely to present at a specialized center cannot be clarified with certainty. Interestingly, Alese and colleagues also described an association between treatment at an academic center and better survival in NEC patients [39]. 

It is difficult to analyze the influence of a single treatment in a heterogeneous cohort, sometimes with multiple treatment lines, on the overall outcome. In a recent retrospective analysis of treatment outcomes of extrapulmonary NEC patients, Mc Garrah et al. described a longer OS in patients treated with cisplatin plus etoposide compared to carboplatin plus etoposide, whereas they could not demonstrate an influence of second-line treatments [54]. In our NEC subgroup, patients treated with first-line cisplatin and etoposide had a non-significant trend toward superior survival as compared to carboplatin plus etoposide (mOS 18 vs. 14 months, *p* = 0.2), whereas the outcome of cisplatin- versus carboplatin-treated patients was very similar (12 and 11 months mOS, respectively) in the NORDIC NEC trial [16]. As information on performance status is missing in the publication of Mc Garrah et al., we cannot exclude that the better results achieved with cisplatin are caused by the selection of patients with better PS for this treatment. A randomized clinical trial is necessary to validate whether cisplatin is superior to carboplatin as first-line treatment for gastroenteropancreatic NEC patients. 

The benefit of surgery in high-grade NEN is unclear. Whereas it is an accepted approach for patients with localized disease [55] with or without (neo-)adjuvant treatment, it is usually not recommended in metastatic disease [1,56]. In our cohort, a high proportion of patients (41.6%) underwent surgery as the first treatment. Surprisingly, this was also true for 29.9% of patients diagnosed with stage IV disease. In a recent analysis of high-grade neuroendocrine tumors in the SEER registry, 27.9% of patients underwent upfront surgery, mainly for localized disease [39]. As in the SEER registry, high-grade is not defined by the proliferation rate (Ki67), and the subgroup of NET G3 patients is not included in this analysis. A very recent report on NET G3 patients [40] documented 40.1% primary tumor resections, which is comparable to our data. The role of surgery may be different for NET G3 and NEC patients [57]. Three retrospective studies recently reported a favorable overall survival despite high recurrence rates in NEN G3 patients with liver metastases treated with curatively intended surgery [58,59,60]. Of course, selection and reporting bias (publication of positive results) has to be taken into consideration. According to the most recent ESMO and NANETS guidelines [19,61], surgery is contraindicated in metastatic NEC patients, whereas it is an option in metastatic NET G3 if R0 resection can be achieved. 

The current WHO classifications separate NET G3 from NEC, with the first introduction of the NET G3 subgroup in pancreatic NEN in 2017. Interestingly, although the majority of patients in our study were diagnosed before 2017, the choice of treatments in German centers has already been influenced by the morphological subtype, as illustrated in Figure 2a–d. This is in accordance with current guidelines [1,19,56] and also supported by a recent publication on outcomes of patients with NET G3 [40]. In this retrospective analysis of treatment outcomes of 142 patients, all other treatments combined resulted in a longer progression-free survival compared to platinum in combination with etoposide (PE). The choice of non-PE versus PE as first-line treatment was the only significant prognostic parameter for PFS in univariate analysis [40]. We were not able to document a significant influence of the first-line treatment in NET G3 patients on mOS, but there was a slight trend toward better survival with the use of STZ + FU (only used in pancreatic NET G3 patients) and temozolomide plus capecitabine as compared to platinum-containing combinations. 

An insufficient response rate to PE in patients with NEN G3 and proliferation rate (Ki67) < 55%, irrespective of the morphological classification, was first reported in the Nordic NEC trial [16]. In the update publication, the response rate to PE was 44% in NEC patients with Ki67 above 55% and only 24% and 25% for NET G3 and NEC patients with Ki67 < 55%, respectively [42]. This challenges the new WHO classification as the basis for the choice of treatment, since the Ki67 value seems to be more informative for predicting responsiveness to PE than the morphological classification.

We also demonstrated that time to treatment failure in first- as well as second-line treatment was shorter with PE as compared to Tem/Cap, STZ/FU, FOLFOX and PRRT in NEN G3 patients with Ki67 values < 55%. 

In the future, an international clinical trial for metastatic NET G3 patients should define a standard first-line treatment by the randomized comparison of Tem/Cap and FOLFOX as often used therapeutic strategies for these patients.

PRRT is usually indicated in progressive, well-differentiated NET G1 and G2 and is best established for midgut NET based on the NETTER-1 trial [62]. Some retrospective studies have also reported favorable results of PRRT for selected patients with metastatic NEN G3 [33,63,64]. In our cohort, a total of 43 patients were treated with PRRT, most frequently as second-line treatment (*n* = 19). Although the number seems insufficient to draw final conclusions, PRRT as second-line treatment performs comparatively well, with a median time to treatment failure of 12 months. In line with previous publications, it seems reasonable to consider PRRT as an option in carefully selected NEN G3 patients and not to exclude this treatment a priori. Two international multicenter randomized clinical trials (NCT03972488 and NCT04919226) will evaluate the potential role of PRRT as first-line treatment in metastatic NET (G2 and) G3 patients. 

As outcomes in some subgroups, such as metastatic colorectal NECs, are still dismal, new treatment concepts are urgently needed. In the French FOLFIRINEC trial (NCT04325425), the investigators will test whether first-line mFOLFIRINOX can prolong mPFS compared to PE [65]. Dual immunotherapy in the DART trial resulted in a promising response rate of 44% in extrapancreatic NEC patients [66], but with the inclusion of only eight GI NEC patients, it is too early to judge the efficacy of dual immunotherapy for this subgroup. Several trials of immunotherapy alone (NCT03095274; NCT03352934; NCT03591731) or in combination with chemotherapy (NCT05058651; NCT03728361) or tyrosine kinase inhibitors (NCT04400474; NCT05015621; NCT05289856) are currently evaluating the potential benefit of immunotherapy for NEC patients. Microsatellite instability, which is known to be associated with response to immunotherapy [67], has been reported in 0–15% of NEC cases [68,69,70]. 

While the separation of NET G3 and NEC as different entities has been supported by molecular analyses demonstrating similar genetic alterations in pancreatic NET G3 and pancreatic NET G1/2 (most often mutations in DAXX, ATRX and MEN) and different alterations in NEC (p53 and Rb1) [71,72,73], data on molecular alterations as predictive biomarkers for the choice of treatment are very limited. Hijioka et al. reported that KRAS mutations and loss of Rb1 were associated with better response to PE in pancreatic NEN G3 [74], which was recently supported by data from a French cohort of NEN G3 of different origin, which demonstrated a significantly higher response to PE in the absence of Rb1 staining [75]. In a very recent publication by the Nordic group, molecular analyses of high-grade NEN with next-generation sequencing (NGS) using a panel of 360 cancer genes revealed a high rate of targetable alterations [70]. BRAF mutations were frequently found in NEC of the colon, and prior case reports have demonstrated responses to BRAF-MEK inhibition in colorectal NEC with the BRAF V600E mutation [76]. In the absence of established effective second-line treatments, the use of NGS could be valuable to define personalized treatment options. We suggest conducting biomarker-driven basket trials to gather efficacy data and to ensure access to targeted drugs, as reimbursement outside clinical trials is often an issue.

Our study has several limitations. Due to the registry approach, a significant subset of the whole cohort presented with incomplete data sets, particularly due to changes in classification criteria, which occurred during the inclusion period. The first data in the German NET Registry were documented retrospectively. The median observation time of 15 months was rather short, and the aim to complete follow-up data with personal visits was hampered by the COVID pandemic. Therefore, there is a wide range of completeness of records on follow-up treatments and their outcomes, and the numbers of individual second- or higher-line treatments in subgroups are low, making it difficult to draw conclusions regarding efficacy. As RECIST criteria are often not used in clinical routine and/or are not documented in the registry, we were not able to analyze response rates or progression-free survival and to compare these data to published series. Nevertheless, using time to treatment failure instead, which also considers changes in treatment strategy due to toxicity or patient wishes, gives some hints on clinically relevant outcome data. Of course, these data cannot replace results from randomized clinical trials with well-defined inclusion criteria and endpoints, which are urgently needed. We tried to reduce heterogeneity by excluding high-grade NEN of the urogenital tract, breast and skin. Nevertheless, this analysis included several different entities with low numbers in some subgroups. The cohort collected from more than 30 centers assures a representative picture of routine patient care within a European health care system, rather than monocentric retrospective data, which are more prone to selection bias, as in the majority of publications on this topic. On the other hand, population-based registries often lack detailed histological information, such as the proliferation rate. A major limitation is the lack of a central reference pathological review. The differentiation between NET G3 and NEC can be difficult in some cases, and criteria for diagnosing NET G3 have recently been described [42]. There is the risk that cases are classified as NEC just by a proliferation rate (Ki67) exceeding 20% while disregarding morphology, especially between 2010 and 2017. A re-evaluation of the available histological material of “the Nordic NEC cohort” [16] by expert pathologists resulted in 12.3% NET G3 cases [42]. The proportion of NET G3 in our registry cohort clearly exceeded this value with 31.7%, making it unlikely that a high number of patients with relatively low proliferation rates were misdiagnosed as NEC, although this cannot be completely ruled out. 

## 5. Conclusions

In this study of a large multicenter cohort of patients diagnosed with high-grade NEN, the median overall survival of 31 months was better than in publications of population-based registries. This may partly be explained by a rather low median proliferation rate of Ki67 of 50%. Nevertheless, in some subgroups, such as metastatic colorectal NEC, the prognosis is still dismal. The best strategy for this subgroup and for NET G3 and NEC with Ki67 < 55%, as well as the choice of an appropriate second-line treatment, remains unclear. Due to the rarity of the subgroups, a joint international effort is needed to conduct clinical trials for well-defined entities. Biomarker-driven basket trials are also desirable. 

High-grade NEN comprises several clinically as well as prognostically different tumors. Therefore, it is time to leave the “one fits all concept” of platinum plus etoposide in all metastatic NEN G3 cases as the first treatment. A more tailored approach is required that takes morphology, proliferation rate, primary tumor localization, somatostatin receptor expression and also molecular markers into consideration.

## Figures and Tables

**Figure 1 cancers-14-02718-f001:**
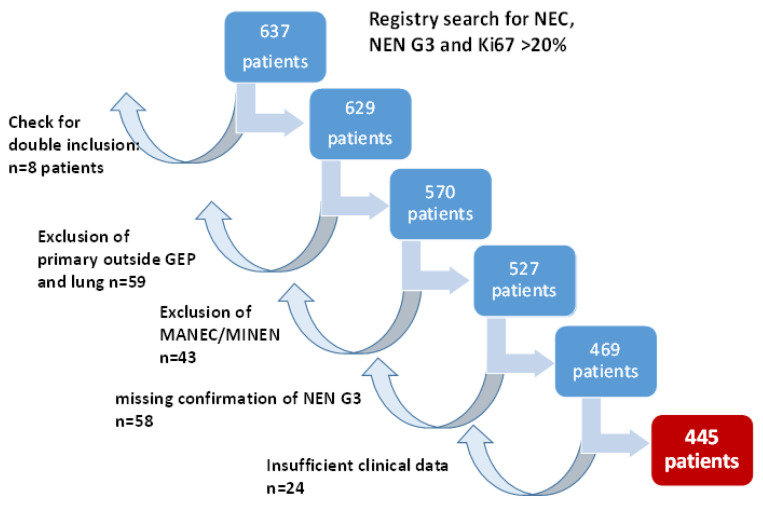
Patient selection for this trial. GEP: gastroenteropancreatic; MANEC: mixed adeno-neuroendocrine carcinoma; MiNEN: mixed neuroendocrine/non-neuroendocrine neoplasms.

**Figure 2 cancers-14-02718-f002:**
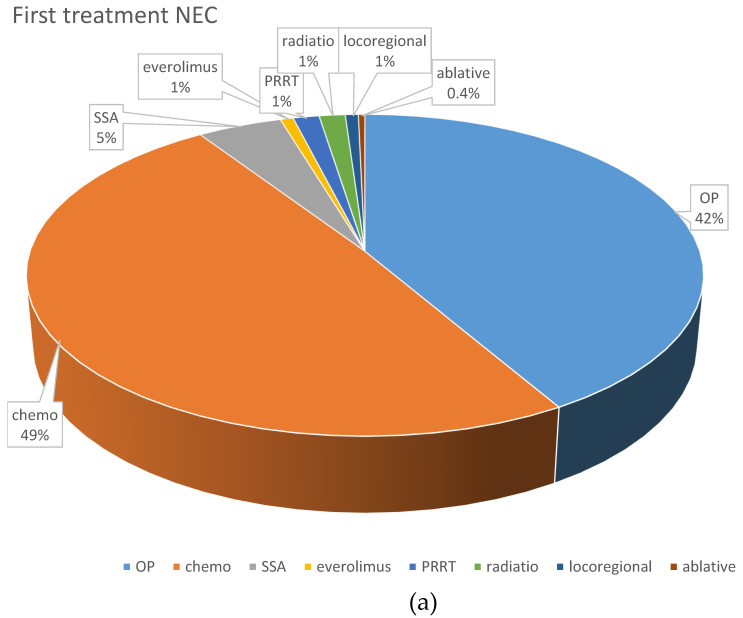
Distribution of first-line treatments in patients classified as NEC or NET. (**a**) First-line NEC, all treatments. (**b**) First-line NET G3, all treatments. (**c**) First-line chemotherapeutic protocols in NEC. (**d**) First-line chemotherapeutic protocols in NET G3. Ablative: local ablative treatment such as radiofrequency ablation or microwave ablation; carbo mono: monotherapy with carboplatin; carbo/tax: carboplatin + paclitaxel; chemo: chemotherapy; DTIC: dacarbazine; drugs nd: chemotherapy protocol not documented; EP: platinum + etoposide; FOLFOX: oxaliplatin + 5-fluorouracil; OP: operative treatment; others: sum of other regimens used in less than 1% of patients; PRRT: peptide receptor radionuclide treatment; SSA: somatostatin analog; STZ/FU: streptozocin+ 5-fluorouracil; Tem/Cap: temozolomide + capecitabine.

**Figure 3 cancers-14-02718-f003:**
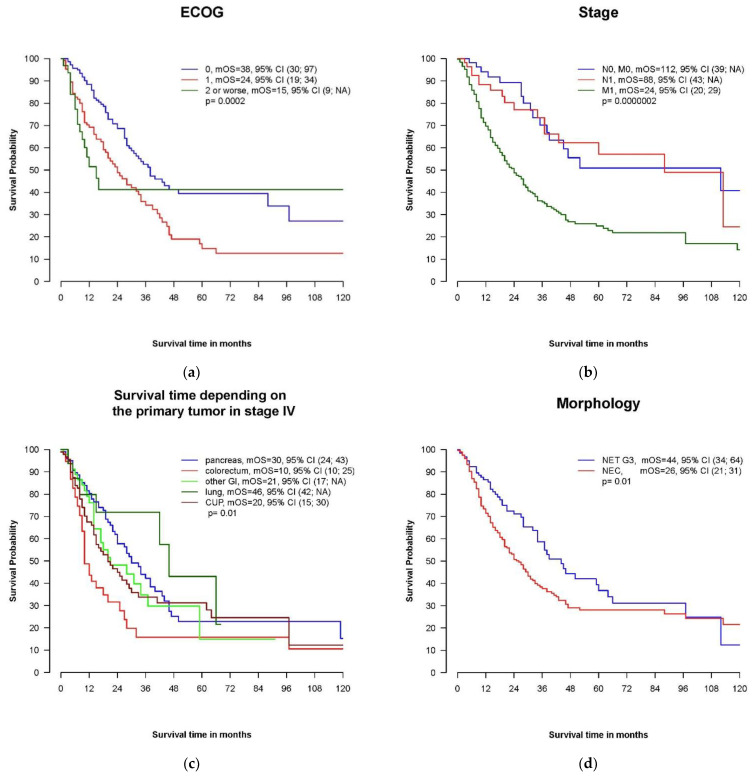
Kaplan–Meier plots of putative prognostic factors. (**a**) Overall survival in patients with ECOG performance statuses of 0, 1 and 2 or worse. ECOG: Eastern Co-operative Oncology Group Performance Status. (**b**): Overall survival depending on stage. (**c**) Overall survival depending on the primary tumor localization in metastatic disease. GI, gastrointestinal; CUP, cancer of unknown primary. (**d**) Overall survival depending on morphology: NEC versus NET. (**e**) Overall survival by proliferation rate: Ki67 ≥ 55% versus Ki67 < 55%. (**f**) Overall survival by proliferation rate: Ki67 ≥ 50% versus Ki67 < 50%. (**g**) Overall survival by serum level of LDH (lactate dehydrogenase). (**h**) Overall survival by plasma level of chromogranin A (CgA).

**Table 1 cancers-14-02718-t001:** Patient characteristics.

Parameter	Number or Median	Percentage or Range
Age at diagnosis (years)	63	18–87
Gender		
Male	246	55.3
Female	199	44.7
ECOG at diagnosis G3		
ECOG 0	149	33.5
ECOG1	127	28.5
ECOG2	28	6.3
ECOG3	4	0.9
ECOG4	1	0.2
Unknown	136	30.6
Stage at diagnosis G3		
I–II	59	13.3
III	59	13.3
IV	318	71.5
Unknown	9	2.0
Location of metastases		
Liver	300	67.4
Lymph nodes ^1^	246	55.3
Bone	95	21.3
Peritoneum	53	11.9
Lung	54	12.1
Brain	38	8.5
Others ^2^	61	13.7
Grading at initial diagnosis		
G1 or G2	53	11.9
G3	392	88.1
Ki67 at diagnosis of G3 (%)		
21–30	143	32.2
31–40	49	11.0
41–50	47	10.6
51–60	24	5.4
61–70	37	8.3
71–80	70	15.8
81–90	53	11.9
91–100	21	4.7
Median Ki67	50%	21–100
Morphology		
NET	141	31.7
NEC	267	60.0
Unknown	37	8.3
Functionality		
Functioning	26	5.8
In NET	13	9.2
In NEC	9	3.4
Unknown	4	10.8
Non-functioning	403	90.1
Unknown	16	3.6
Primary tumor localization		
Pancreas	132	29.7
Colorectal	85	19.1
Gastrointestinal, others *	86	19.3
Lung	45	10.1
• Atypical carcinoid/NET	24	5.3
• LCNEC	18	4.0
• Unknown	3	0.7
CUP	97	21.8

NET: neuroendocrine tumor; NEC: neuroendocrine carcinoma; CUP: cancer of unknown primary; ECOG: Eastern Co-operative Oncology Group Performance Status; LCNEC: large cell neuroendocrine carcinoma; * other gastrointestinal includes esophagus, stomach, duodenum, jejunum, ileum and appendix; ^1^ number includes locoregional as well as distant lymph node metastases; ^2^ other metastatic sites include adrenal, mesenteric, heart, ovarian, muscle, pancreatic, subcutaneous, pleural, renal, gastric, breast, pericardial, soft tissue and splenic metastases.

**Table 2 cancers-14-02718-t002:** Multivariate Analysis.

Parameter	HR	95% CI	*p* Value
Stage IV	4.006	2.54–6.33	<0.0001
Ki67 (continuous)	1.025	1.018–1.032	<0.0001
ECOG PS ≥ 1	2.231	1.594–3.124	<0.0001

HR: hazard ratio; CI: confidence interval; ECOG PS: Eastern Cooperative Oncology Group Performance Status, available in 309 cases.

## Data Availability

The data presented in this study are available on request from the corresponding author. The data are not publicly available due to privacy and ethical reasons.

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
