# Peer review of "Treatment Approaches and Outcome of Patients with Neuroendocrine Neoplasia Grade 3 in German Real-World Clinical Practice"

_cancers, 2022, doi:10.3390/cancers14112718_

Round 1

Reviewer 1 Report

This work deals with assessment of the real world data of group of the German NET Registry i.e. patients suffering neuroendocrine tumours.

The aim of authors is to enhance efforts toward conducting clinical trials with respect to diagnostic and prognostic features.

The treatment outcomes and demographic date were taken into consideration.

I think the work is exhaustive in case of describing the workflow for data collection, treatment and evaluation.

In conclusion, please, provide the future aims of authors in this scope of investigation.

Author Response

We are very grateful to the reviewer for taking the time and giving valuable comments.

This work deals with assessment of the real world data of group of the German NET Registry i.e. patients suffering neuroendocrine tumours.

The aim of authors is to enhance efforts toward conducting clinical trials with respect to diagnostic and prognostic features.

We agree with this assessment.  We hope that these analyses of real world data and treatment outcome are a first step to stimulate the development of clinical trials as progress in the field of NEN G3 is an unmet medical need.

The treatment outcomes and demographic date were taken into consideration.

Thanks for appreciation.

I think the work is exhaustive in case of describing the workflow for data collection, treatment and evaluation.

Thank you. We tried to give detailed information.

In conclusion, please, provide the future aims of authors in this scope of investigation.

There are two major future aims of the authors in the field of NEN G3: 1. Find appropriate therapeutic options for NET G3 and "low proliferative" NEC- situations without defined standard of care. We are involved in 2 German phase II trials, one is now added on page 20, and hope with international joint effort a randomized trial in metastatic NET G3 could be conducted (also added on page 20). 2. For NEC patients especially biomarker driven basket trials could be a valuable option as added on page 21. These aspects were already briefly mentioned in "Conclusions".

Reviewer 2 Report

Wide series of G3 NET patients undergoing different treatment approaches. 

Main limitation is the retrospective design. I am afraid the analysis could not be able to account for all the potential confounders.

Other main limitation is the follow-up length. In fact, NET studies require a very long follow-up period given the long survival observed with these neoplasms.

Finally, i would suggest to expand in the discussion the comment on the role of surgery (of the primary tumor) in patients with NET, also citing the paper PMID: 27956320. 

Author Response

We are very thankful for the comments of the reviewer and the appreciation of our manuscript. Please find below a point-to-point reply.

Wide series of G3 NET patients undergoing different treatment approaches. 

Yes, we report on a large cohort of patients with NEN G3 (NET as well as NEC) with different treatment approaches as a result of the analysis of the German NET Registry data.

Main limitation is the retrospective design. I am afraid the analysis could not be able to account for all the potential confounders.

We totally agree that a retrospective design of a study is always a major limitation. As described in the section "Patients and Methods" the data are mainly collected prospectively (after 2004), so we have a retrospective analysis of prospectively collected data. This limitation is mentioned in the discussion on page 21.  As also pointed out we had to deal with incomplete data sets due to the registry approach. Therefore, unfortunately,  a consideration of all possible confounders was not possible.

Other main limitation is the follow-up length. In fact, NET studies require a very long follow-up period given the long survival observed with these neoplasms.

We fully agree that for patients with NET G1 and NET G2 - the more common NET patients with good prognosis - a long follow up is required.  Our cohort comprises only NEN G3 cases (NET G3 and NEC) with clearly less favorable outcome. This is also demonstrated by the fact that 210 of 445 included patients died during the observation period. Therefore the number of events allowed prognostic analyses. Nevertheless, a longer duration of follow-up would be desirable and we mentioned the rather short median follow up of 15 months as a limitation on page 21 in the discussion.

Finally, i would suggest to expand in the discussion the comment on the role of surgery (of the primary tumor) in patients with NET, also citing the paper PMID: 27956320. 

Thank you for this suggestion. In this interesting publication Citterio and colleagues analysed the role of primary tumor resection in patients with metastatic functioning NET G1 / NET G2. NEN G3 was an exclusion criterion. As we only included patients with NEN G3 in our study this paper is outside the focus of our manuscript. A broader discussion of the role of surgery in metastatic NET could be an interesting topic for a review - as there are some publications available- but goes beyond the scope of our paper. Therefore, we only included publications on surgery in the G3 situation.